# The relationship between noise annoyance, emotional labor and burnout in operating-room nurses: A protocol for cross-sectional study

Yizhi Zhang[1,2☯], Yan Chen[3☯], Jiaqing Rao[2], Qingjiao Guo[4*], Jing Ou-yang[2], Shasha Luo[2], Ying Gu[2*]

1 Operating Room, Beijing Jishuitan Hospital Guizhou Hospital, Guiyang, Guizhou, China, 2 Department of Nursing, Affiliated Hospital of Guizhou Medical University, Guiyang, Guizhou, China, 3 Department of Urology, GUIQIAN International General Hospital, Guiyang, Guizhou, China, 4 Department of Nursing, Hospital of Stomatology of Zunyi, Zunyi, Guizhou, China

☯ These authors contributed equally to this work.
* 1014348500@qq.com (YG); 1726110302@qq.com (QG)

## Abstract

Operating room nurses may experience occupational burnout brought by noise annoyance during which emotional labor may act as a moderating factor between the two. The aims of this study are: (1) to investigate the independent correlation between noise annoyance and burnout among operating room nurses; (2) to explore the relationship between noise annoyance, emotional labor, and burnout. This study, a cross-sectional study design, is being conducted in Guizhou province, China. The Chinese version of the noise annoyance, emotional labor and burnout scale will be used to assess the current situation of the samples, and the data will be collected using paper questionnaires and electronic questionnaires. Additionally, the Pearson or Spearman rank correlation coefficients will be involved in correlation analyses. Multiple linear regression analysis will be also carried out to produce β-coefficients with 95% confidence intervals. A nonparametric, bias-corrected 95% bootstrapped confidence interval with 5,000 bootstrap iterations will help to determine the importance of mediating effects. The trial registration number is ChiCTR2300068414.

## Introduction

Burnout represents a significant challenge to the well-being of nurses and their professional environment [1]. Defined as a prolonged response to chronic interpersonal and emotional stressors in the workplace, burnout is a complex occupational psychosocial syndrome [2]. In mainland China, the prevalence of burnout among registered nurses is alarmingly high, reaching up to 50% [3]. Operating Room (OR) nurses, due to the high-intensity, high-risk, and overload of their work, are particularly vulnerable to burnout [4]. A study conducted by Ren Lanzhu reported that 78.77% of OR nurses experience varying degrees of burnout [5]. The implications of burnout extend

**Data availability statement:** The data underlying the pilot survey presented in this study are available in the figshare repository at https://figshare.com/articles/dataset/____/25466992?-file=45245002, with the DOI 10.6084/m9.figshare.25466992. And all subsequent relevant data from this study will be made available upon study completion.

**Funding:** The authors received no specific funding for this work.

**Competing interests:** The authors have declared that no competing interests exist.

beyond individual well-being, affecting nurses' physical and mental health, increasing turnover intentions, and negatively impacting care quality, patient safety, and job satisfaction [6]. Moreover, research indicates that burnout among nurses correlates with higher patient mortality rates and increased hospital-acquired infections [7]. Therefore, exploring the factors contributing to burnout in OR nurses and identifying effective intervention strategies are crucial for improving their mental health.

Noise annoyance (NA) is defined as the adverse emotional response of individuals or groups exposed to noisy environments. It manifests through feelings of distraction, discomfort, irritation, and anxiety, serving as one of the most significant psychological impacts of noise [8,9]. Noise annoyance is widely regarded as a key indicator of adverse subjective reactions to noise. OR nurses are likely to experience heightened levels of noise annoyance due to their work environment. A study revealed that 95.2% of anesthesiologists working in ORs reported annoyance caused by noise, with 14.5% experiencing high levels of noise annoyance [10]. Furthermore, burnout is closely linked to negative emotional states [11]. Noise annoyance, an often overlooked negative emotion [12], may be significantly associated with burnout. However, the precise relationship between noise annoyance and burnout remains underexplored, warranting further investigation.

Emotional labor (EL) refers to the process by which individuals regulate their emotions to align with organizational expectations, resulting in specific facial expressions and behavioral displays required by their roles [13]. OR nurses frequently engage in emotional labor to provide patient care [14]. Bartram confirmed that emotional labor is a significant positive predictor of burnout [15]. And negative emotions are one of the factors influencing emotional labor [16]. Since noise annoyance is a form of negative emotion [12], the relationship between noise annoyance and emotional labor in OR nurses has yet to be explored.

While the relationship between emotional labor and burnout appears well-established, the link between noise annoyance and burnout remains unclear, and the role of emotional labor in the connection between noise annoyance and burnout is still unknown. Therefore, the objectives of this study are: (1) to investigate the independent correlation between noise annoyance and burnout among OR nurses; (2) to explore the relationship between noise annoyance, emotional labor, and burnout.

### Research hypothesis

As early as 1988, Topf identified noise pollution as a significant stressor in the nursing work environment, with noise-induced stress serving as a predictive factor for burnout among clinical nurses [17]. Noise annoyance, as a specific form of psychological stress [18], may influence the onset of burnout, particularly given the well-established link between stress and burnout [19]. In addition, a survey of 2,000 nurses in Taiwan examining burnout identified feelings of tension, irritability, sadness, inferiority, and sleep disturbances as the most significant predictors of burnout [20]. Compared to factors such as job satisfaction, job engagement, and work environment, psychological factors were found to have the most substantial correlation with emotional exhaustion [13,14]. Negative emotions have also been shown to

correlate positively with burnout [11]. From a psychological perspective, noise annoyance in OR nurses represents a form of negative emotion [18], which may, in turn, contribute to burnout.

The Conservation of Resources theory posits that the persistent threat of resource loss can lead to burnout, ultimately impacting patient safety [19]. This theory is grounded in the premise that individuals strive to acquire, maintain, and protect resources they value, providing a useful framework for understanding burnout. It highlights the psychological stress experienced by workers in human services, including healthcare and education, as a result of interpersonal demands [21,22]. For OR nurses, exposure to noise annoyance can deplete emotional resources, and if these resources are not replenished in a timely manner, further depletion may lead to burnout.

Grandey identified several factors influencing emotional labor, including emotional intelligence, emotional expression, emotional disposition, and emotional events (both positive and negative) [23]. And negative emotions are significantly correlated with emotional labor [16], which is considered a key predictor of burnout [24]. The Self-Regulation Theory refers to the process by which individuals control or alter their thoughts, emotions, impulses, and behaviors to achieve specific goals [25]. In the context of emotional labor, OR nurses adjust and display appropriate emotions to meet work requirements or objectives, exemplifying a typical self-regulation process. Before engaging in emotional labor for patients, OR nurses may already be experiencing varying degrees of noise annoyance. The higher the level of noise annoyance, the more pronounced the emotional dysregulation, which increases the likelihood of emotional exhaustion and ultimately leads to burnout. Furthermore, when OR nurses engage in emotional labor, noise annoyance forces them to exert more self-regulation, thereby increasing the demand for emotional resources. When emotional demands and resources are significantly out of balance, burnout is likely to accelerate.

Based on these considerations, this study proposes the following hypotheses:

a) There is a positive correlation between noise annoyance and burnout among OR nurses.

b) There is a positive correlation between noise annoyance and emotional labor among OR nurses.

c) There is a positive correlation between emotional labor and burnout among OR nurses.

d) Emotional labor mediates the relationship between noise annoyance and burnout among OR nurses.

The hypothesized model for this study is illustrated in Fig 1.

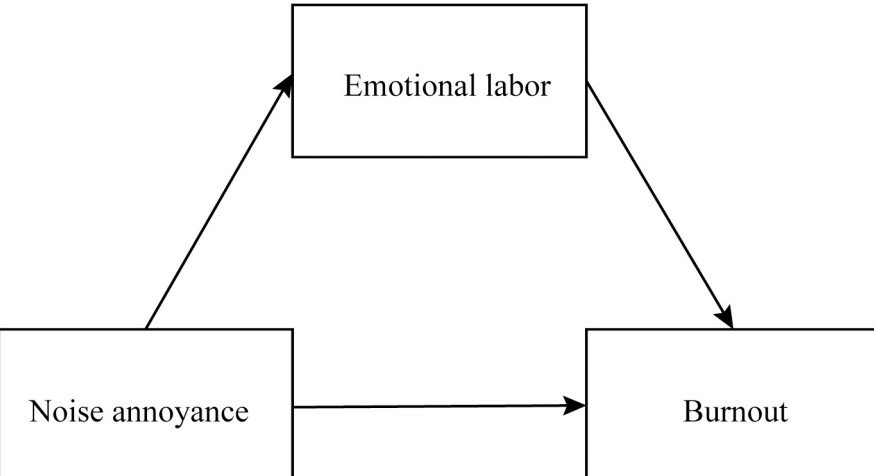

**Fig 1. The conceptual framework of the relationship between noise annoyance, emotional labor and burnout.**

## Materials and methods

### Study design and setting

We will conduct this descriptive, cross-sectional study on a convenience sample in ORs of 11 hospitals in Guizhou Province, China. These hospitals are all tertiary hospital with more than 500 beds.

### Participants

Participants will be recruited from 11 hospitals in accordance with the criteria below. The inclusion criteria are: (1) Registered nurses who have worked in ORs for more than 1 year; (2) They should work in the corresponding post in ORs during the investigation period; (3) They provide informed consents and voluntary participation in this study. Visiting fellowship nurses and practical nurses will be excluded in our study.

The required sample size was calculated using G*Power 3.1.9.7 software [26] for multiple linear regression analysis, with four predictor variables: noise annoyance, surface acting, deep acting, and natural emotional expression, and burnout as the outcome variable. While emotional labor is our primary construct of interest, it is operationalized through three distinct dimensions—surface acting, deep acting, and natural emotional expression—each serving as an individual predictor. This differentiation allows for a more nuanced examination of how each aspect of emotional labor contributes to burnout among OR nurses.

The test family used was F-tests, and the statistical test employed was linear multiple regression (Fixed model, $R^2$ deviation from zero). A priori: Compute required sample size - given α, power, and effect size, was used for the type of power analysis. Assuming a medium effect size ($f^2 = 0.15$), an alpha level of 0.05, and a desired power of 0.95, the minimum required sample size was determined to be 129 participants. To account for a potential 20% dropout rate, the final minimum sample size was calculated to be 155. This sample size ensures sufficient power to detect significant relationships within our multiple regression model [27].

### Variables

The demographics of participants include sex, age, and nationality. Personal factors consist of marital status, education level, personality traits, and health status. Work-related factors such as work experience as an OR nurse, professional title, annual income, and hospital tenure are also considered. Additional variables used to describe the sample are satisfaction with salary (does not meet expectations, meet expectations, or far exceed expectations) and average hours of noise exposure per day. The item "What personality traits do you think you belong to?" will be used for personality traits assessment and the answers include stability, emotionality, extroversion, introversion, and psychosis [28]. Furthermore, a self-reported assessment of the population's health status will be made as "not very healthy" (suffering from severe disease), "fair" (having chronic disease), "healthy" and "very healthy".

The noise annoyance level of the subjects will be measured by the descriptive rating scale recommended by GB/Z 21233-2007 called "Acoustics—Assessment of noise annoyance by means of social and socio-acoustics surveys" [29], which is the Chinese version of the International Organization for Standardization Technical Specification (ISO/TS) 15666 (2003) [30]. The item is: "Thinking about the last month or so, when you were here working, how much did the noise from the operating room bother, disturb, or annoy you" (not at all = 1, slightly = 2, moderately = 3, very = 4, or extremely = 5)? According to the description of the factors that cause noise in "AORN Position Statement on Managing Distractions and Noise During Perioperative Patient Care" [31], we will measure the annoyance caused by five noise sources, including equipment operations and troubleshooting, behavioral activities, metal instruments and movable equipment of ORs to OR nurses. We consider an overall average score of 4 and above as high annoyance [10]. The measure's Cronbach's alpha value was 0.947 in pilot survey of this study.

The Emotional Labor Scale (ELS) will adopt the three-dimensional structural scale of emotional labor proposed by Chinese scholar Luo Hong based on Grandey's version [32], combined with the actual situation of Chinese clinical nursing,

and it has reasonable reliability when used in nurse groups. The scale includes three dimensionality: surface acting (7 items), deep acting (3 items), and natural emotional expression (4 items), with a total of 14 items. The questionnaire's Cronbach's alpha value for the three dimensions are 0.711, 0.826 and 0.872, respectively [33], and 0.838, 0.858, and 0.713 respectively in pilot survey of this current study. A 6-point scoring method is used, and from strongly disagree to strongly agree is given 1–6 points in turn; the higher the score, the more times the individual feels emotional adjustment. The total average score of items <2 is considered low-level emotional labor, 2–4 is considered medium-level emotional labor, and >4 is considered high-level emotional labor.

This study will use the MBI-HSS (MBI Human Services Survey) scale translated and revised by scholars at the Hong Kong Polytechnic University for the nursing population [34]. The Cronbach's alpha coefficients for each dimension of the questionnaire are between 0.80 and 0.85, and 0.950, 0.923, and 0.953 respectively in pilot survey of our research. The scale has a total of 22 items and 3 dimensions (emotional exhaustion, depersonalization, and personal accomplishment). The evaluation adopts a 7-point Likert scale, with "never", "rarely", "occasionally", "often", "frequently", "very frequently", and "daily" scoring 0–6 points, respectively. We define that a score above or equal to 27 on the emotional exhaustion dimension, above or equal to 8 on the depersonalization dimension, or below or equal to 24 on the personal fulfillment dimension is positive [35]. Additionally, the scores of the three dimensions all lower than the threshold value are considered to be without burnout; the score of one dimension higher than the threshold value is viewed as mild burnout; the score of each of the two dimensions higher than the threshold value is viewed as moderate burnout; and the scores of the three dimensions higher than the critical value is viewed as severe burnout [35]. Besides, the total incidence rate of OR nurse burnout = [(number of people with mild burnout + number of people with moderate burnout + number of people with severe burnout)/ total number of people] × 100% [36]. Since burnout is measured on a continuous scale, we consider that a higher score reflects more severe symptoms of burnout [37].

## Measurement

Between February 18 and March 17, 2023, a pilot survey was conducted on 30 nurses in OR of the Affiliated Hospital of Guizhou Medical University by research assistants, and questionnaires were filled out on site. Based on the pilot test results, the content of the questionnaire was adjusted to form a final version. Subsequently, we started a formal investigation on March 18th, 2023. Officials from Guizhou Medical University, along with the director of the Nursing Department of the Affiliated Hospital of Guizhou Medical University contacted the head nurse in OR of the target hospitals, and research assistants explained the purpose and relevance of the study to them. The head nurse of the OR of the target hospital recruited nurses who meet the inclusion criteria. After obtaining respondents' verbal consent for data collection, they were issued an electronic questionnaire and an electronic informed consent form. In addition, nurses who completed the questionnaire were be paid 2 RMB for their time and effort. Data collection was scheduled to the end of July 2023, but the exact time depended on the progress of the research.

In this study, the research assistants were trained and qualified nursing master's students. The training content included the types, structures, question formulations, response types and survey methods of questionnaires by professional courses.

## Bias control

We tried to control for non-response bias by explaining the purpose and value of the study to respondents through phone or email, as people are more likely to respond to the questionnaire when they understand the reason for the survey. Due to the fact that participants' past experience of noise annoyance and burnout at any time could lead to recall bias, we asked respondents to relate to events that occurred within the past month when filling out the questionnaire. To reduce selection bias to some extent, we chose hospitals in a certain region of China (Guizhou Province) as the research site,

and all hospitals are tertiary hospitals. Since all the participants come from tertiary hospitals in the same region, they have basically the same social culture and similar workloads. This makes the results comparable.

## Statistical analysis

We will review the returned questionnaires one by one to check whether they are filled in completely. If the data has obvious regularity or consistency (for example, if you select the same option in the entire questionnaire), the questionnaire will be considered invalid [38], and we will delete it. Excel 2016 will be used to manage data, and IBM SPSS Statistics 26.0 (IBM Corp., 2019) will be used for statistical analysis. For describing the data, the quantitative data conforming to the normal distribution will be represented by the mean and standard deviation after calculating the skewness and kurtosis of the data. Quantitative data that does not conform to the normal distribution will be expressed as medians and interquartile ranges. The comparison of noise annoyance, emotional labor and burnout scores of OR nurses with different general demographic data will be performed according to the results of the homogeneity of variance test. If the variance is homogeneous, we will use independent sample student's t-test and one-way ANOVA; if not, we will use non-parametric tests (Wilcoxon rank sum test and Kruskal-Wallis H test). In addition, qualitative data will be presented as frequencies and percentages. Pearson correlation coefficient or Spearman correlation coefficient (according to the results of the normality test) will be conducted to explore the correlation between variables. For the correlation coefficient, 0 means there is no correlation between the two, -1 means negative correlation, and 1 means positive correlation. The closer it is to -1 or 1, the stronger the correlation [38].

Potential confounders as contributors to burnout, and possibly related to noise annoyance and emotional labor, were defined a priori and incorporated into the statistical model, such as sex, age, marital status, education level, personality traits, health status, work experience as an operating room nurse, professional title, annual income, satisfaction with salary and hospital tenure. We will carry out linear multiple regression analysis to control for confounders and determine whether noise annoyance and emotional labor are contributors to burnout. Before performing multiple regression analyses, we will examine scatterplots of each predictor variable against the burnout score to check for linear relationships. The normality of residuals will be tested using the Shapiro-Wilk test and visualized with Q-Q plots. And we will evaluate homoscedasticity by plotting residuals versus predicted values. The Durbin-Watson statistic will be used to test for autocorrelation of residuals. In case non-linear relationships are identified, we will consider applying transformations to the predictor variables or incorporating polynomial or interaction terms into the regression model to better capture the nature of the relationships.

To address the risk of multicollinearity arising from multiple confounders, we will calculate Variance Inflation Factors (VIF) and tolerance levels for each predictor variable. A VIF value exceeding 5 or a tolerance value below 0.2 will indicate significant multicollinearity [39]. If multicollinearity is detected, we will consider removing highly correlated variables.

We will use Model 4 (PROCESS macro) in SPSS 26.0 to examine the mediation effect of emotional labor on the relationship between noise annoyance and burnout among OR nurses [40]. A nonparametric, bias-corrected 95% bootstrapped confidence interval (BCI) with 5,000 bootstrap iterations will be used to determine the importance of mediating effects. Indirect effects will be deemed significant if the 95% BCI do not contain zero [41,42]. A p-value of 0.05 or lower will be regarded as statistically significant, based on a two-sided test.

## Ethical considerations

We obtained approval from the corresponding institutional review board (2022 Ethics Review No. 746). The questionnaire contained the following information about the participants: research purpose, and guarantees of anonymity, voluntary participation, and that their personal data would be used only for the purpose of this research. In addition, participants were informed that they had the right to withdraw from the survey at any time without giving reasons and without any negative consequences. At last, all participants provided written informed consent for participation.

## Strengths and limitations of this study

• This is the first survey in China to verify whether noise annoyance is an important factor affecting burnout.

• It is also the first study in China to examine whether emotional labor is a mediator between noise annoyance and burnout.

• A limitation of this study is that personality traits of participants are not measured using a thorough valid and reliable instrument since this is not the main objective of this study. In future exploration of the impact of personality traits on burnout in more detail, we will use more comprehensive and valid measurement scales.

• Another limitation is that the cross-sectional study design cannot determine causality of variables.

• In addition, our use of self-reported measures can also be considered a study limitation, which may introduce a risk of social desirability bias.

## Author contributions

**Conceptualization:** Yizhi Zhang, Yan Chen.

**Methodology:** Jiaqing Rao, Qingjiao Guo.

**Supervision:** Jing Ou-yang, Ying Gu.

**Validation:** Jiaqing Rao, Shasha Luo.

**Visualization:** Jiaqing Rao, Qingjiao Guo.

**Writing – original draft:** Yizhi Zhang.

**Writing – review & editing:** Yan Chen, Jing Ou-yang, Ying Gu.

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
