## [Decision Letter · Decision Letter 0]

9 Feb 2024

PONE-D-23-14792The relationship of noise annoyance, emotional labor and burnout in operating room nurses: protocol for a cross-sectional studyPLOS ONE

Dear Dr. Zhang,

Thank you for submitting your manuscript to PLOS ONE. After careful consideration, we feel that it has merit but does not fully meet PLOS ONE’s publication criteria as it currently stands. Therefore, we invite you to submit a revised version of the manuscript that addresses the points raised during the review process.

We look forward to receiving your revised manuscript.

Kind regards,

Sally Mohammed Farghaly

Academic Editor

PLOS ONE

Journal Requirements:

Reviewers' comments:

Reviewer's Responses to Questions

**Comments to the Author**

1. Is the manuscript technically sound, and do the data support the conclusions?

Reviewer #1: Partly

2. Has the statistical analysis been performed appropriately and rigorously? 

Reviewer #1: N/A

3. Have the authors made all data underlying the findings in their manuscript fully available?

Reviewer #1: No

4. Is the manuscript presented in an intelligible fashion and written in standard English?

Reviewer #1: No

5. Review Comments to the Author

Reviewer #1: Manuscript number: PONE-D-23-14792

Title: The relationship of noise annoyance, emotional labor and burnout in operating room

nurses: protocol for a cross-sectional study

Objectives: (1) to assess the significant influencing factors of burnout among OR nurses; (2) to explore the relationship among noise annoyance, emotional labor, and burnout

Overall, this is a clearly written study protocol. However, I have some major and minor comments to this protocol:

Background:

1. The line: “In addition, noise …exposure to noise[6,7]” is a bit strange to me because how is noise annoyance accompanied by cognitive assessment?

2. It is stated that, compared to ward nurses OR nurses should provide patients greater compassionate care and emotional support. This needs a more in depth explanation and references.

Materials and Methods:

3. Please describe more clearly when and how the data are collected. Additionally, also how participants are recruited (advertisement, via the wards, social media, etc.).

4. Add (more) information about the reliability and validity of the used questionnaires and instruments (including references).

5. Please explain the sentence “… whether authorized personnel will be considered.”

6. The average hours of noise exposure per day are collected. However, is this by participant report? This might have a major recall bias. Moreover, when is sound regarded as ‘noise’?

7. The authors ask the participants : “what personality traits do you think you belong to?” Is this based on a thorough valid and reliable personality test? If so provide this information. If not provide information why this question can replace a personality test.

8. He sentence “the study will … nurse translation.” is unclear. Please rephrase.

9. The authors define a score in the MBI-HSS above a certain level as positive. Please provide references and/or a thorough explanation why the authors think this is a methodologically sound way to do this.

10. Sorry, but I can’t follow the sentence “Additionally, … total x 100%. Please make this more clear.

11. In the calculation the mild, moderate and severe burnout are divided by ‘total’. What does this ‘total’ mean? Please explain.

12. In the measurements and Bias section, is stated that the researchers will explain the purpose and significance of the survey. Shouldn’t this be the purpose and relevance?

13. Nurses who completed the questionnaire will be paid for their time and effort. State how much money this is. Moreover, have the authors thoughts how the minimize the bias this might give?

14. What do the authors mean by: ”the baselines are basically close?

15. In the statistical analysis is stated that data that has “obvious regularity and consistency” will be deleted. What does this mean? How is this addressed? And what methods are used for this check?

16. Pease add information about the outcomes of the spearman and person correlations. When is it by example a weak or moderate correlation?

17. In the potential confounders ‘whether authorized personnel’ is stated again. Please explain this.

Patient and public involvement:

18. It is stated that participants will be interviewed. How will this be done? And why is this not described in the method section? Please add this to the method section

19. In addition, who are the trained research assistants? How are they trained, what is their professional background? Are they involved in the study? Please add this to the method section

6. PLOS authors have the option to publish the peer review history of their article (what does this mean? ). If published, this will include your full peer review and any attached files.

**Do you want your identity to be public for this peer review?** For information about this choice, including consent withdrawal, please see our Privacy Policy .

Reviewer #1: **Yes: ** Hans Timmerman, PhD.

---

## [Author Response · Author response to Decision Letter 0]

25 Mar 2024

Dear editor and reviewer:

Thank you for handling our submission and offering us an opportunity of revision, as well as providing us with valuable comments concerning our manuscript entitled “The relationship of noise annoyance, emotional labor and burnout in operating room nurses: protocol for a cross-sectional study” (ID: PONE-D-23-14792). Those comments are helpful for revising and improving our paper, as well as the important guiding significance to our research. We have studied the comments carefully and have addressed all the requirements and comments, and detailed all the changed made to the manuscript which we hope meet with approval.

Journal Requirements:

Journal Requirement 1: Please ensure that your manuscript meets PLOS ONE's style requirements, including those for file naming.

Authors’ response: Thank you very much for your honest advice. We have contacted a professional organization to make changes according to PLOS ONE's style requirements, which we hope will meet with approval.

Journal Requirement 2: We note that you have indicated that there are restrictions to data sharing for this study. PLOS only allows data to be available upon request if there are legal or ethical restrictions on sharing data publicly.

Authors’ response: We are very sorry for any inconvenience caused to the journal. Since this study is a research protocol, it is currently in progress and the data collection has not yet been completed. However, we have uploaded the pilot survey data set of this study with a DOI of 10.6084/m9.figshare.25466992, as the pilot survey has been completed at this stage. And relevant data from this study will all be made available upon study completion. The following explanations have been added in the revised manuscript, which are replicated here for your easy reference:

Data Availability Statement

The data underlying the pilot survey presented in the study are available from a DOI of 10.6084/m9.figshare.25466992. All subsequent relevant data from this study will be made available upon study completion. (Page 15, Line 314-317)

Journal Requirement 3: Your ethics statement should only appear in the Methods section of your manuscript. If your ethics statement is written in any section besides the Methods, please move it to the Methods section and delete it from any other section. Please ensure that your ethics statement is included in your manuscript, as the ethics statement entered into the online submission form will not be published alongside your manuscript.

Authors’ response: Thank you very much for your practical advice, based on your request we have deleted the ethics statement written in another section besides the Methods.

Reviewers' comments

Overall response to Reviewer: Thank you very much for your positive response and kind suggestions on our work. We have endeavored to incorporate the feedback and have revised the manuscript accordingly. The detailed point-by-point responses are listed below. All changes made to the manuscript are marked in red so that they may be easily identified.

Reviewers' comment 1: The line: “In addition, noise …exposure to noise [6,7]” is a bit strange to me because how is noise annoyance accompanied by cognitive assessment?

Authors’ response: For the first comment, thank you very much for your suggestion, in response to which we have again reviewed and found that there was an error in the expression here. Our true intention was to express that noise annoyance can lead to impairments in cognitive assessment, and we have corrected this. We believe the added explanation help to address your concerns. The following sentence has been modified in the revised manuscript, which is replicated here for your easy reference:

In addition, noise annoyance is often accompanied by negative emotional responses such as restlessness, distress, exhaustion, and cognitive assessment impairment, as well as a lack of control over exposure to noise [6,7]. (Page 3, Line 47-50)

Reviewers' comment 2: It is stated that, compared to ward nurses OR nurses should provide patients greater compassionate care and emotional support. This needs a more in depth explanation and references.

Authors’ response: Thank you so much for your suggestion. We understand you are concerned about this statement, which is missing a more in-depth explanation and references. Regarding this concern, we have added more explanations in the revised manuscript as follows:

For patients, being transferred from the ward to the operating room means they enter a relatively closed and unfamiliar environment, and be separated from their family members causes psychological discomforts to them. Due to preoperative restrictions on food and water, other factors such as poor rest at night before surgery also bring physical discomforts to them. In addition, surgical treatment contributes both physical and psychological traumas to patients to a certain extent, which increases patients' preoperative stress and anxiety. As a result, operating-room nurses need to provide more compassionate care and emotional support to patients [11]. (Page 3, Line 59-66)

Reviewers' comment 3: Please describe more clearly when and how the data are collected. Additionally, also how participants are recruited (advertisement, via the wards, social media, etc.).

Authors’ response: We are grateful for your suggestion that we should make more detail about data collection. To illustrate this issue more clearly, we include measurement as a separate paragraph. And we have provided a more detailed description of the data collection and participant recruitment methods in the “Measurement” section. We believe the added explanation help to address your concerns. The following explanations have been added in the revised manuscript, which are replicated here for your easy reference:

Between February 18 and March 17, 2023, a pilot survey was conducted on 30 nurses in OR of the Affiliated Hospital of ** Medical University by research assistants, and questionnaires were filled out on site. Based on the pilot test results, the content of the questionnaire was adjusted to form a final version. Subsequently, we started a formal investigation on March 18th, 2023. Officials from ** Medical University, along with the director of the Nursing Department of the Affiliated Hospital of ** Medical University contacted the head nurse in OR of the target hospitals, and research assistants explained the purpose and relevance of the study to them. The head nurse of the OR of the target hospital recruited nurses who meet the inclusion criteria. After obtaining respondents' verbal consent for data collection, they were issued an electronic questionnaire and an electronic informed consent form. In addition, nurses who completed the questionnaire were be paid 2 RMB for their time and effort. Data collection was scheduled to the end of July 2023, but the exact time depended on the progress of the research. (Page 10, Line 212-225)

Reviewers' comment 4: Add (more) information about the reliability and validity of the used questionnaires and instruments (including references).

Authors’ response: We are grateful for your suggestion, and we understand that valid data comes from reliable questionnaires. Since the pilot survey of this study has been completed, we calculated the reliability of the questionnaires based on the results of the pilot survey. We believe the added explanation help to address your concerns. The following information have been added in the revised manuscript, which are replicated here for your easy reference:

The noise annoyance level of the subjects will be measured by the descriptive rating scale recommended by …average score of 4 and above as high annoyance [36]. The measure’s Cronbach’s alpha value was 0.947 in pilot survey of this study. (Page 8-9, Line 164-177)

The Emotional Labor Scale (ELS) will adopt the three-dimensional … in nurse groups. The questionnaire’s Cronbach's alpha value for the three dimensions are 0.711, 0.826 and 0.872, respectively [38], and 0.838, 0.858, and 0.713 respectively in pilot survey of this current study. (Page 9, Line 178-183)

This study will use the MBI-HSS (MBI Human Services Survey) scale … nursing population [39]. The Cronbach's alpha coefficients for each dimension of the questionnaire are between 0.800 and 0.850 in a Chinese study [39], and 0.950, 0.923, and 0.953 in pilot survey of our research. (Page 9-10, Line 191-195)

Reviewers' comment 5: Please explain the sentence “… whether authorized personnel will be considered.”

Authors’ response: Thank you for your comment, and we are very sorry for any inconvenience this caused. “Whether authorized personnel” means an official public position in China. We have changed it to hospital tenure, a more international language. The specific modifications are as follows:

… Work-related factors such as work experience as an OR nurse, professional title, annual income, and hospital tenure are also considered. (Page 8, Line 154-156)

Potential confounders as contributors to burnout…, satisfaction with salary and hospital tenure. (Page 13, Line 267)

Reviewers' comment 6: The average hours of noise exposure per day are collected. However, is this by participant report? This might have a major recall bias. Moreover, when is sound regarded as ‘noise’?

Authors’ response: Thank you for your comment. This is one of the limitations of this study since its nature. First, due to the difficulty of working shifts, we are unable to measure the duration of noise exposure for each nurse, so this study chooses the form of participant self-report for data collection. Secondly, individuals have different perceptions and sensitivities to noise, which may cause bias in the noise annoyance and the duration of exposure to noise reported by research subjects [1, 2], but these variables are internal experiences that are best reported by individuals themselves. Consequently, we believe that self-reports are suitable for measuring such variables. And we will conduct further targeted investigation in the future. We hope this explanation helps to address your concerns. The following information have been added in the revised manuscript, which are replicated here for your easy reference:

A limitation of this study is that … In addition, our use of self-reported measures can also be considered a study limitation, which may introduce a risk of social desirability bias. (Page 14, Line 295-296)

Reviewers' comment 7: The authors ask the participants: “what personality traits do you think you belong to?” Is this based on a thorough valid and reliable personality test? If so provide this information. If not provide information why this question can replace a personality test.

Authors’ response: Thank you for your comment. This is also one of the limitations of this study. The purpose of this study is to explore the relationship between noise annoyance, emotional labor and burnout. The relationship between personality traits and burnout is not the focus of this investigation. In addition, three main scales were involved in this study, with a total of 41 items. If a standard scale is used to describe personality traits, there will be too many items, which will affect the quality of the data [3]. Because as far as we know, the comprehensively valid and reliable personality trait test scales in China have between 15 and 48 items [4, 5]. Therefore, we used this single item to collect data on participants' personality traits, described in demographic variables. In future exploration of the impact of personality traits on burnout in more detail, we will use more comprehensive and valid measurement scales. We hope this explanation helps to address your concerns. The following information have been added in the revised manuscript, which are replicated here for your easy reference:

A limitation of this study is that personality traits of participants are not be measured using a thorough valid and reliable instrument since this is not the main objective of this study. In future exploration of the impact of personality traits on burnout in more detail, we will use more comprehensive and valid measurement scales. (Page 14, Line 289-292)

Reviewers' comment 8: He sentence “the study will … nurse translation.” is unclear. Please rephrase.

Authors’ response: We are very grateful for your suggestion, and we are sorry for any inconvenience this caused. Since this sentence is indeed unclear, we have rephrased it. The following sentence has been modified in the revised manuscript, which are replicated here for your easy reference:

This study will use the MBI-HSS (MBI Human Services Survey) scale translated and revised by scholars at the Hong Kong Polytechnic University for the nursing population [39]. (Page 9, Line 191-193)

Reviewers' comment 9: The authors define a score in the MBI-HSS above a certain level as positive. Please provide references and/or a thorough explanation why the authors think this is a methodologically sound way to do this.

Authors’ response: We are very grateful for your suggestion, and we have provided references. The following information have been added in the revised manuscript, which are replicated here for your easy reference:

We define that a score above or equal to 27 on the emotional exhaustion dimension, above or equal to 8 on the depersonalization dimension, or below or equal to 24 on the personal fulfillment dimension is positive [40]. Additionally, the scores of the three dimensions all lower than the threshold value are considered to be without burnout; the score of one dimension higher than the threshold value is viewed as mild burnout; the score of each of the two dimensions higher than the threshold value is viewed as moderate burnout; and the scores of the three dimensions higher than the critical value is viewed as severe burnout [40]. Besides, the total incidence rate of OR nurse burnout = (number of people with mild burnout + number of people with moderate burnout + number of people with severe burnout) / total number of people × 100% [41]. (Page 10, Line 199-209)

Reviewers' comment 10: Sorry, but I can’t follow the sentence “Additionally, … total x 100%. Please make this more clear.

Authors’ response: We are very sorry for the inconvenience of reading this and thank you very much for your suggestion, we have amended the sentence as follows, which we hope will help you understand our research better.

Besides, the total incidence rate of OR nurse burnout = (number of people with mild burnout + number of people with moderate burnout + number of people with severe burnout) / total number of people × 100%. (Page 10, Line 206-209)

Reviewers' comment 11: In the calculation the mild, moderate and severe burnout are divided by ‘total’. What does this ‘total’ mean? Please explain.

Authors’ response: Thank you so much for your comment. The “total” refers to the “total number of people”. We are deeply sorry for the unclear expression. We hope this explanation helps to address your concerns.

Reviewers' comment 12: In the measurements and Bias section, is stated that the researchers will explain the purpose and significance of the survey. Shouldn’t this be the purpose and relevance?

Authors’ response: We are very grateful for your suggestion, and we have made the modification in our manuscript. The following sentence has been modified in the revised manuscript, which are replicated here for your easy reference:

Between February 18 and March 17, 2023, a pilot survey was conducted … contacted the head nurse in OR of the target hospitals, and research assistants explained the purpose and relevance of the study to them. (Page 11, Line 219)

Reviewers' comment 13: Nurses who completed the questionnaire will be paid for their time and effort. State how much money this is. Moreover, have the authors thoughts how the minimize the bias this might give?

Authors’ response: We are very grateful for your suggestion. The following explanation have been added in the revised manuscript, which are replicated here for your easy reference:

Between February 18 and March 17, 2023, a pilot survey was conducted … In addition, nurses who completed the questionnaire were be paid 2 RMB for their time and effo

---

## [Decision Letter · Decision Letter 1]

21 Oct 2024

PONE-D-23-14792R1The relationship of noise annoyance, emotional labor and burnout in operating room nurses: protocol for a cross-sectional studyPLOS ONE

Dear Dr. Zhang,

Thank you for submitting your manuscript to PLOS ONE. After careful consideration, we feel that it has merit but does not fully meet PLOS ONE’s publication criteria as it currently stands. Therefore, we invite you to submit a revised version of the manuscript that addresses the points raised during the review process.

We look forward to receiving your revised manuscript.

Kind regards,

Sally Mohammed Farghaly

Academic Editor

PLOS ONE

Reviewers' comments:

Reviewer's Responses to Questions

**Comments to the Author**

1. If the authors have adequately addressed your comments raised in a previous round of review and you feel that this manuscript is now acceptable for publication, you may indicate that here to bypass the “Comments to the Author” section, enter your conflict of interest statement in the “Confidential to Editor” section, and submit your "Accept" recommendation.

Reviewer #2: All comments have been addressed

Reviewer #3: (No Response)

Reviewer #4: (No Response)

2. Is the manuscript technically sound, and do the data support the conclusions?

Reviewer #2: Yes

Reviewer #3: No

Reviewer #4: No

3. Has the statistical analysis been performed appropriately and rigorously? 

Reviewer #2: No

Reviewer #3: No

Reviewer #4: No

4. Have the authors made all data underlying the findings in their manuscript fully available?

Reviewer #2: No

Reviewer #3: Yes

Reviewer #4: Yes

5. Is the manuscript presented in an intelligible fashion and written in standard English?

Reviewer #2: Yes

Reviewer #3: Yes

Reviewer #4: Yes

6. Review Comments to the Author

Reviewer #2: In this manuscript, all comments have been addressed. But there still are some things that could be improved.

The research topic of this manuscript is: The relationship of noise annoyance, emotional labor and burnout in operating room nurses. This manuscript contains some statements that are not relevant to the research topic. For example, surgical treatment contributes both physical and psychological traumas to patients to a certain extent, which increases patients' preoperative stress and anxiety. operating-room nurses need to provide more compassionate care and emotional support to patients [11]. Then, the author mentioned emotional labor. The context is incoherent. There are logical problems with the content of this manuscript. The content requires major revision. In addition, the authors set four predictor variables when estimating the minimum sample size. But according to the authors' research structure there are only two predictor variables. What are the other two variables? According to the research hypothesis. This manuscript does not require independent samples t-test and variance analysis. Why should the author consider control variables? What is the theoretical basis? It is recommended that the author seek assistance from statistical experts. From the manuscript's content, I am concerned that the statistical analysis of this manuscript is wrong. Finally, the research structure of this manuscript is logically correct. However, the textual narrative lacks logic. This manuscript still requires major revision.

Reviewer #3: Line 133: The exact name is to be stated if the manuscript is accepted for publication.

Line 143: For the sample size calculation, the outcome variable needs to be specified. For GPower, indicate the test family, the statistical test, and the type of power analysis used.

Line 204-206: Use brackets to present the formula, ensuring that 100% is clearly separated from the denominator.

Line 214: Replace the word ‘effectively’ with a more appropriate alternative.

Line 244: Proper citation of the SPSS including the publisher's name is to be stated.

Line 253: Replace the word ‘parametric’ with ‘non-parametric’.

Line 256: Specify that the p-value is based on a two-sided test.

Line 260-269: More details could be provided for this section.

Line 266-268: Clearly indicate whether the analysis will be conducted using the SPSS PROCESS macro or another method.

Line 270: The word ‘significant’ is to be replaced with ‘statistically significant’.

The fulfillment of multiple regression assumptions should be stated. If there are non-linear relationships between variables, the results may not accurately reflect the true mediation effect.

When controlling for multiple confounders, there is a risk of multicollinearity, which can inflate standard errors and make it difficult to assess the unique contribution of each variable. This needs to be addressed.

Line 303: Why was PLOS mentioned here? Does PLOS play a role in this study?

Line 300 and 303: These sentences need revision.

Line 314: Provide the access weblink.

Line 318: The sentence is unclear and needs revision.

Single-item measure is generally inadequate for capturing the personality traits and could lead to inaccurate/incomplete assessments even if it is not the primary focus/objective of the study.

References: Some references do not conform to the journal’s format and the references is to be thoroughly checked.

Reviewer #4: Line 144-146 Sample size: how many nurses are anticipated to have burn out? Is this the 5%. Will logistic regression be used? This needs to be understandable to which regression will be done, for which sort of outcome and predictor.

The stats analysis section reads as if it may be linear regression if normally distributed outcome variable of burnout score. State this clearly in the analysis section and the sample size (the method to assess the primary outcome should be used in the sample size justification.

7. PLOS authors have the option to publish the peer review history of their article (what does this mean? ). If published, this will include your full peer review and any attached files.

**Do you want your identity to be public for this peer review?** For information about this choice, including consent withdrawal, please see our Privacy Policy .

Reviewer #2: **Yes: ** De Chih, Lee

Reviewer #3: No

Reviewer #4: No

---

## [Author Response · Author response to Decision Letter 1]

8 Dec 2024

Response Letter

Dear Editor and Reviewers:

Thank you for handling our submission and offering us an opportunity of revision, as well as providing us with valuable comments concerning our manuscript entitled “The relationship of noise annoyance, emotional labor and burnout in operating room nurses: protocol for a cross-sectional study” (ID: PONE-D-23-14792). Those comments are helpful for revising and improving our paper, as well as the important guiding significance to our research. We have studied the comments carefully and have addressed all the requirements and comments, and detailed all the changed made to the manuscript which we hope meet with approval.

Reviewers' comments

Overall response to Reviewer: Thank you very much for your positive response and kind suggestions on our work. We have endeavored to incorporate the feedback and have revised the manuscript accordingly. The detailed point-by-point responses are listed below.

Reviewer #2

Comment 1: The research topic of this manuscript is: The relationship of noise annoyance, emotional labor and burnout in operating room nurses. This manuscript contains some statements that are not relevant to the research topic. For example, surgical treatment contributes both physical and psychological traumas to patients to a certain extent, which increases patients' preoperative stress and anxiety. operating-room nurses need to provide more compassionate care and emotional support to patients [11]. Then, the author mentioned emotional labor. The context is incoherent. There are logical problems with the content of this manuscript. The content requires major revision.

Authors’ response: Thank you very much for your comment. Your suggestion have truly helped us improve the quality of the paper. Based on your recommendation, we have removed irrelevant statements and made major revisions to the content. We believe the revisions help to address your concerns. The following sentences have been modified in the revised manuscript, and are replicated here for your easy reference:

Introduction

Burnout represents a significant challenge to the well-being of nurses and their professional environment [1]. Defined as a prolonged response to chronic interpersonal and emotional stressors in the workplace, burnout is a complex occupational psychosocial syndrome [2]. In mainland China, the prevalence of burnout among registered nurses is alarmingly high, reaching up to 50% [3]. Operating Room (OR) nurses, due to the high-intensity, high-risk, and overload of their work, are particularly vulnerable to burnout [4]. A study conducted by Ren Lanzhu reported that 78.77% of OR nurses experience varying degrees of burnout [5]. The implications of burnout extend beyond individual well-being, affecting nurses’ physical and mental health, increasing turnover intentions, and negatively impacting care quality, patient safety, and job satisfaction [6]. Moreover, research indicates that burnout among nurses correlates with higher patient mortality rates and increased hospital-acquired infections [7]. Therefore, exploring the factors contributing to burnout in OR nurses and identifying effective intervention strategies are crucial for improving their mental health.

Noise annoyance (NA) is defined as the adverse emotional response of individuals or groups exposed to noisy environments. It manifests through feelings of distraction, discomfort, irritation, and anxiety, serving as one of the most significant psychological impacts of noise [8,9]。Noise annoyance is widely regarded as a key indicator of adverse subjective reactions to noise. OR nurses are likely to experience heightened levels of noise annoyance due to their work environment. A study revealed that 95.2% of anesthesiologists working in ORs reported annoyance caused by noise, with 14.5% experiencing high levels of noise annoyance [10]. Furthermore, burnout is closely linked to negative emotional states [11]. Noise annoyance, an often overlooked negative emotion [12], may be significantly associated with burnout. However, the precise relationship between noise annoyance and burnout remains underexplored, warranting further investigation.

Emotional labor (EL) refers to the process by which individuals regulate their emotions to align with organizational expectations, resulting in specific facial expressions and behavioral displays required by their roles [13]. OR nurses frequently engage in emotional labor to provide patient care [14]. Bartram confirmed that emotional labor is a significant positive predictor of burnout[15]. And negative emotions are one of the factors influencing emotional labor [16]. Since noise annoyance is a form of negative emotion [12], the relationship between noise annoyance and emotional labor in OR nurses has yet to be explored.

While the relationship between emotional labor and burnout appears well-established, the link between noise annoyance and burnout remains unclear, and the role of emotional labor in the connection between noise annoyance and burnout is still unknown. Therefore, the objectives of this study are: (1) to investigate the independent correlation between noise annoyance and burnout among OR nurses; (2) to explore the relationship between noise annoyance, emotional labor, and burnout.

Research hypothesis

As early as 1988, Topf identified noise pollution as a significant stressor in the nursing work environment, with noise-induced stress serving as a predictive factor for burnout among clinical nurses [17]. Noise annoyance, as a specific form of psychological stress [18], may influence the onset of burnout, particularly given the well-established link between stress and burnout [19]. In addition, a survey of 2,000 nurses in Taiwan examining burnout identified feelings of tension, irritability, sadness, inferiority, and sleep disturbances as the most significant predictors of burnout [20]. Compared to factors such as job satisfaction, job engagement, and work environment, psychological factors were found to have the most substantial correlation with emotional exhaustion [13,14]. Negative emotions have also been shown to correlate positively with burnout[11]. From a psychological perspective, noise annoyance in OR nurses represents a form of negative emotion[18], which may, in turn, contribute to burnout.

The Conservation of Resources theory posits that the persistent threat of resource loss can lead to burnout, ultimately impacting patient safety [19]. This theory is grounded in the premise that individuals strive to acquire, maintain, and protect resources they value, providing a useful framework for understanding burnout. It highlights the psychological stress experienced by workers in human services, including healthcare and education, as a result of interpersonal demands [21,22]. For OR nurses, exposure to noise annoyance can deplete emotional resources, and if these resources are not replenished in a timely manner, further depletion may lead to burnout.

Grandey identified several factors influencing emotional labor, including emotional intelligence, emotional expression, emotional disposition, and emotional events (both positive and negative) [23]. And negative emotions are significantly correlated with emotional labor [16], which is considered a key predictor of burnout [24]. The Self-Regulation Theory refers to the process by which individuals control or alter their thoughts, emotions, impulses, and behaviors to achieve specific goals [25]. In the context of emotional labor, OR nurses adjust and display appropriate emotions to meet work requirements or objectives, exemplifying a typical self-regulation process. Before engaging in emotional labor for patients, OR nurses may already be experiencing varying degrees of noise annoyance. The higher the level of noise annoyance, the more pronounced the emotional dysregulation, which increases the likelihood of emotional exhaustion and ultimately leads to burnout. Furthermore, when OR nurses engage in emotional labor, noise annoyance forces them to exert more self-regulation, thereby increasing the demand for emotional resources. When emotional demands and resources are significantly out of balance, burnout is likely to accelerate.

Based on these considerations, this study proposes the following hypotheses:

a) There is a positive correlation between noise annoyance and burnout among OR nurses.

b) There is a positive correlation between noise annoyance and emotional labor among OR nurses.

c) There is a positive correlation between emotional labor and burnout among OR nurses.

d) Emotional labor mediates the relationship between noise annoyance and burnout among OR nurses.

The hypothesized model for this study is illustrated in Fig 1.

Comment 2: In addition, the authors set four predictor variables when estimating the minimum sample size. But according to the authors' research structure there are only two predictor variables. What are the other two variables?

Authors’ response: Thank you for highlighting the discrepancy regarding the number of predictor variables used in our sample size estimation. We appreciate the opportunity to clarify this aspect of our study. In our research, while we primarily focus on two main predictor constructs—noise annoyance and emotional labor—the construct of emotional labor is further subdivided into three distinct dimensions: surface acting, deep acting, and natural emotional expression. Each of these dimensions represents a unique aspect of emotional labor and is treated as a separate predictor variable in our multiple regression analysis. Therefore, in total, we have four predictor variables: Noise Annoyance, Surface Acting, Deep Acting, and Natural Emotional Expression. This approach allows us to explore the nuanced effects of each dimension of emotional labor on burnout, providing a more comprehensive understanding of the factors contributing to burnout among operating room nurses. We hope the explanation helps to address your concerns. And we have made the revisions as follows:

The required sample size was calculated using G*Power 3.1.9.7 software [26] for multiple linear regression analysis, with four predictor variables: noise annoyance, surface acting, deep acting, and natural emotional expression, and burnout as the outcome variable. While emotional labor is our primary construct of interest, it is operationalized through three distinct dimensions—surface acting, deep acting, and natural emotional expression—each serving as an individual predictor. This differentiation allows for a more nuanced examination of how each aspect of emotional labor contributes to burnout among OR nurses.

Comment 3: According to the research hypothesis. This manuscript does not require independent samples t-test and variance analysis. Why should the author consider control variables? What is the theoretical basis? It is recommended that the author seek assistance from statistical experts. From the manuscript's content, I am concerned that the statistical analysis of this manuscript is wrong.

Authors’ response: Thank you for your valuable comment. We appreciate the opportunity to clarify the rationale behind our methodological approach.

Justification for Including Control Variables:

Our primary objective is to investigate the independent association between noise annoyance and burnout among operating room nurses. Burnout is a complex, multifactorial phenomenon influenced by various personal and professional factors [1, 2]. To accurately assess the unique effect of noise annoyance on burnout, it is essential to control for potential confounding variables that may also impact burnout levels.

Theoretical Basis:

The inclusion of control variables is grounded in the Conservation of Resources Theory Model which mentioned in the INTRODUCTION. According to this model, burnout develops when job demands are high and resources are insufficient. Personal characteristics (e.g., age, marital status, education level, personality traits) and job-related factors (e.g., work experience, professional title, income, job satisfaction) can influence both the perception of noise annoyance and the experience of burnout [1, 2].

By controlling for these variables, we aim to isolate the specific effect of noise annoyance on burnout, ensuring that our findings are not confounded by other factors known to influence burnout.

Use of Independent Samples t-test and ANOVA (Univariate Analysis):

Prior to conducting multiple regression analysis, we perform univariate analyses (independent samples t-tests and one-way ANOVA) to identify potential confounders. This step helps us determine which demographic and professional variables are significantly associated with burnout and should be included as control variables in the regression model.

We hope the explanations help to address your concerns.

Comment 4: Finally, the research structure of this manuscript is logically correct. However, the textual narrative lacks logic. This manuscript still requires major revision.

Authors’ response: Thank you so much for your comment. Your suggestion have truly helped us improve the quality of the paper. We have made major revisions to the manuscript and enhanced the logical flow of the text. Due to space limitations, we kindly ask the reviewer to refer back to the manuscript. Thank you once again.

Reviewer #3

Comment 1:

Line 133: The exact name is to be stated if the manuscript is accepted for publication.

Line 143: For the sample size calculation, the outcome variable needs to be specified. For G*Power, indicate the test family, the statistical test, and the type of power analysis used.

Line 204-206: Use brackets to present the formula, ensuring that 100% is clearly separated from the denominator.

Line 214: Replace the word ‘effectively’ with a more appropriate alternative.

Line 244: Proper citation of the SPSS including the publisher's name is to be stated.

Line 253: Replace the word ‘parametric’ with ‘non-parametric’.

Line 256: Specify that the p-value is based on a two-sided test.

Line 266-268: Clearly indicate whether the analysis will be conducted using the SPSS PROCESS macro or another method.

Line 270: The word ‘significant’ is to be replaced with ‘statistically significant’.

Line 314: Provide the access weblink.

Line 318: The sentence is unclear and needs revision.

Authors’ response: Thank you very much for your valuable comments. Your suggestions have truly helped us improve the quality of the manuscript. We have made revisions based on your comments and have replicated here for your easy reference:

The test family used was F-tests, and the statistical test employed was linear multiple regression (Fixed model, R² deviation from zero). A priori: Compute required sample size - given α, power, and effect size, was used for the type of power analysis. Assuming a medium effect size (f² = 0.15), an alpha level of 0.05, and a desired power of 0.95, the minimum required sample size was determined to be 129 participants. To account for a potential 20% dropout rate, the final minimum sample size was calculated to be 155. This sample size ensures sufficient power to detect significant relationships within our multiple regression model [27].

Besides, the total incidence rate of OR nurse burnout =[ (number of people with mild burnout + number of people with moderate burnout + number of people with severe burnout) / total number of people] × 100%

We will review the returned questionnaires one by one to check whether they are filled in completely.

IBM SPSS Statistics 26.0 (IBM Corp., 2019) will be used for statistical analysis.

If the variance is homogeneous, we will use independent sample student’s t-test and one-way ANOVA; if not, we will use non-parametric tests (Wilcoxon rank sum test and Kruskal-Wallis H test).

A p-value of 0.05 or lower will be regarded as statistically significant, based on a two-sided test.

We will use Model 4 (PROCESS macro) in SPSS 26.0 to examine the mediation effect of emotional labor on the relationship between noise annoyance and burnout among OR nurses [40].

The data underlying the pilot survey presented in this study are available at https://figshare.com/articles/dataset/____/25466992?file=45245002, with the DOI 10.6084/

---

## [Decision Letter · Decision Letter 2]

12 Mar 2025

The relationship of noise annoyance, emotional labor and burnout in operating room nurses: protocol for a cross-sectional study

PONE-D-23-14792R2

Dear Dr. Zhang,

We’re pleased to inform you that your manuscript has been judged scientifically suitable for publication and will be formally accepted for publication once it meets all outstanding technical requirements.

Kind regards,

Gholamheidar Teimori-Boghsani

Academic Editor

PLOS ONE

Additional Editor Comments (optional):

Reviewers' comments:

Reviewer's Responses to Questions

**Comments to the Author**

1. Does the manuscript provide a valid rationale for the proposed study, with clearly identified and justified research questions?

Reviewer #3: Partly

Reviewer #4: Yes

2. Is the protocol technically sound and planned in a manner that will lead to a meaningful outcome and allow testing the stated hypotheses?

Reviewer #3: Partly

Reviewer #4: Yes

3. Is the methodology feasible and described in sufficient detail to allow the work to be replicable?

Reviewer #3: Yes

Reviewer #4: Yes

4. Have the authors described where all data underlying the findings will be made available when the study is complete?

Reviewer #3: Yes

Reviewer #4: Yes

5. Is the manuscript presented in an intelligible fashion and written in standard English?

Reviewer #3: Yes

Reviewer #4: Yes

6. Review Comments to the Author

You may also provide optional suggestions and comments to authors that they might find helpful in planning their study.

Reviewer #3: The authors have put in great efforts to address the comments.

No further comments. The manuscript is acceptable for publication.

Reviewer #4: Thank you for the revisions to the manuscript. It is clearer now, particularly the detail about the continuous primary endpoint - although mentioning the incidence may still need the definition of mild, moderate and severe from the continuous scale.

7. PLOS authors have the option to publish the peer review history of their article (what does this mean? ). If published, this will include your full peer review and any attached files.

**Do you want your identity to be public for this peer review?** For information about this choice, including consent withdrawal, please see our Privacy Policy .

Reviewer #3: No

Reviewer #4: No

---

## [Editor Report · Acceptance letter]

PONE-D-23-14792R2

PLOS ONE

Dear Dr. Zhang,

I'm pleased to inform you that your manuscript has been deemed suitable for publication in PLOS ONE. Congratulations! Your manuscript is now being handed over to our production team.

Kind regards,

on behalf of

Dr. Gholamheidar Teimori-Boghsani

Academic Editor

PLOS ONE